# Different geomorphic processes control suspended sediment and bedload export from glaciers

Ian Delaney ①[1] ✉, Frédéric Lardet[1,2], Matt Jenkin ①[1], Davide Mancini[1] & Stuart N. Lane ①[1]

Ongoing cryospheric change has modified sediment export from glacierized catchments substantially, with significant implications for ecosystems and downstream users, notably hydropower companies. Sediment is exported either as finer sediment in suspension or as coarser bedload with intermittent contact between sediment and the bed. To date, the difficulty in observing subglacial bedload transport limits the understanding of the physical processes associated with evacuating bedload compared with suspended load. We elucidate the factors controlling sediment export by inverting a physically-based numerical model of subglacial sediment production and sediment transport with suspended sediment and continuous bedload discharge records from an Alpine glacier. Comparable quantities of suspended sediment and bedload are exported, and model results suggest that both rely on the availability of sediment for transport. Yet, bedload export in subglacial channels also depends on particular hydraulic conditions, notably channel shape and hydraulic roughness. This makes exporting bedload-sized particles inefficient compared to fine-grained sediment. As a result, subglacial hydraulics should be explicitly considered when examining bedload export processes, and suspended and bedload transport should be considered separately. Inefficient bedload evacuation by melt water implies that glacial erosion may only continue when non-fluvial mechanisms evacuate sediment, such as sediment entrainment into the ice.

Establishing the processes driving subglacial erosion and sediment export from glaciers is key as glaciers are extremely effective erosive agents as compared with other mechanisms[1-3], shaping mountain landscapes[e.g.4-7] and driving sediment delivery that impacts hydropower and ecosystems[e.g.8-10]. Underneath glaciers, sediment export responds to the availability of sediment, controlled by glacial erosion processes such as abrasion of bedrock by sliding, debris-laden ice and quarrying of bedrock[1,2,11,12]. Additionally, sediment export responds to subglacial fluvial transport conditions, controlled by the shear stress of water flowing through a subglacial channel with changing size[13-16].

Sediment is exported fluvially either as finer sediment in suspension or as coarser bedload with intermittent contact between sediment clasts and the channel's bed. These two mechanisms have been extensively evaluated in river systems[17,18], where sediment discharge by both processes is a function of sediment supply, sediment size, sediment sorting, and flow conditions. Yet, the sediment supply and transport processes controlling bedload and suspended sediment

[1]Institut des Dynamique de la Surface Terrestre (IDYST), Université de Lausanne Bâtiment Géopolis, CH-1015 Lausanne, Switzerland. [2]Present address: Laboratory of Catchment Hydrology and Geomorphology, School of Architecture, Civil and Environmental Engineering, Ecole Polytechnique Fédérale de Lausanne, Route des Ronquos 86, CH-1951 Sion, Switzerland. ✉e-mail: IanArburua.Delaney@unil.ch

**Fig. 1 | Numerical experiments. a** Cartoon of subglacial processes considered in the model, with inverted parameters noted sediment production rate ($\dot{\varepsilon}$), the sediment height initial condition ($H_0$), the channel shape ($\beta$), and channel roughness ($f_i$); channel cross section modified from Delaney et al.[54] **b** Bedload and suspended load in a subglacial channel with ice above. **c** Map of Otemma glacier with the star denoting measurement station location and blue outline of glacier. Image from SwissTopo.

export in subglacial fluvial systems remain highly uncertain[19]. Continuous high-resolution records of suspended sediment export from glaciers have been extensively collected[e.g.20–23]. Some mechanisms by which glaciers evacuate suspended sediment are reasonably well established from observations[20,24–30]. However, this is not the case for bedload, with the few records available commonly collected some distance downstream from the glacier[31,32], where they are impacted by subaerial hydraulic conditions and proglacial processes that modify significantly the glacier export signal[33,34]. Recent advances in environmental seismology[e.g.35] however, allow researchers to measure bedload below glaciers and at their termini, quantifying the role of bedload transport in a glacier's total sediment export[33]. This means we can now quantify both suspended load and bedload export continuously from glaciers, isolating subglacial sediment transport processes.

Observations show that bedload can comprise up to 50 % of total sediment export from a glacier during a melt season[24,33], and models suggest that glacier quarrying creates bedload-sized material in amounts comparable to the suspended sediment produced by glacier abrasion[36]. The current assumption is that subglacial rivers have the excess capacity necessary to evacuate most of the sediment produced by erosion[e.g.2,19]. However, research has yet to establish the differing hydraulic and geomorphic conditions leading to the fluvial mobilization of quarried-bedload material and abraded-suspended material. It appears that subglacial water can easily mobilize suspended sediment[2,22]. As a result, the mobilization of sediment can depend largely on sediment production and access of subglacial meltwater to abraded sediment[27,30,37,38]. Transporting larger bedload clasts, on the other hand, requires high shear stresses between subglacial water and the conduit bed, and this competence needs to be maintained along the full length of the channel[2,19,39]. As access to glacier beds is limited, subglacial sediment transport processes remains poorly evaluated, including their variability in time and space[40,41]. This knowledge gap inhibits a better understanding of landscape evolution and forecasting sediment discharge from glacierized catchments as they respond to changing climate with differing hydrology and glacial conditions.

Here, we demonstrate quantitatively that different processes control bedload and suspended load export from glaciers, using records collected directly at the glacier terminus. We apply a lumped-element model of glacial erosion and sediment transport in subglacial pressurized flow conditions to topography and continuous hydrology records from the Otemma glacier in Switzerland for the melt seasons of 2020 and 2021 (Fig. 1; see Methods). Otemma glacier is a 6 km long glacier with unknown hard or soft bed characteristics, in an Alpine climate. The model setup is identical for both suspended load (*SSL*) and bedload (*BL*), except for the sediment grain size $D_{50}$ (*SSL*: 0.2 cm ; *BL*: 7.8 cm [33]) and sediment transport relationship (*SSL*: sand-sized sediment transport formula from Engelund and Hansen, 1967[42] and *BL*: bedload transport formula Wilcock and Crowe, 2003[43]). Comparing model outputs to observed sediment transport records in a Monte Carlo framework allows us to evaluate the different parameters controlling sediment availability and subglacial sediment transport conditions. The model application first demonstrates that different subglacial sediment availability and hydraulic processes control suspended and bedload export. Using both forward model outputs and the posterior parameter distributions, we then infer the specific processes controlling bedload and suspended load export. Lastly, we discuss the impact of the different sediment transport processes on landscape evolution in glaciated terrain and on sediment export under current climate warming and glacier retreat.

## Results

### Different hydro-geomorphic processes control bedload and suspended sediment export from glaciers

Model outputs show that different processes control *SSL* and *BL* export from Otemma glacier (Fig. 2). As substantial quantities of *SSL* and *BL* are exported from the glacier[33], this suggests that different parameterizations of the two processes are needed to fully capture subglacial sediment export. The mismatch in processes occurs despite the model being able to successfully represent *SSL* process interactions consistently in 2020 and 2021, along with capturing *SSL* and *BL* variations (Fig. 3).

To evaluate the processes that control *SSL* and *BL*, the model was run with a large number of randomly selected parameter values using inversion to identify values that best represented each transport mode. Parameters control the sediment production rate during the model run ($\dot{\varepsilon}$), the sediment stored below the glacier before the model initialization ($H_0$), the channel shape ($\beta$), and channel roughness ($f_i$; Fig. 1). Both the observations and resultant model outputs were averaged over 5 day periods and compared to the *SSL* and *BL* data using their relative error (see Methods; Fig. 2). Following the inversion, no parameter combinations successfully captured the observed variations in both *BL* and *SSL* (Fig. 2). The failure of the model to reconcile *SSL* and *BL* export together strongly suggests that different physical processes control these two transport modes.

While no single set of parameters accurately represents both *SSL* and *BL* simultaneously (Fig. 2), the model adequately represents

*SSL* and *BL* individually (Fig. 3), with specific parameter combinations. We ran the model 2, 000, 000 times and retained the top performing 0.05% of model runs (1000) for bedload (*BL*) and suspended load (*SSL*) over the years 2020 and 2021, resulting in four different parameter sets. Results from the inversion show that, with

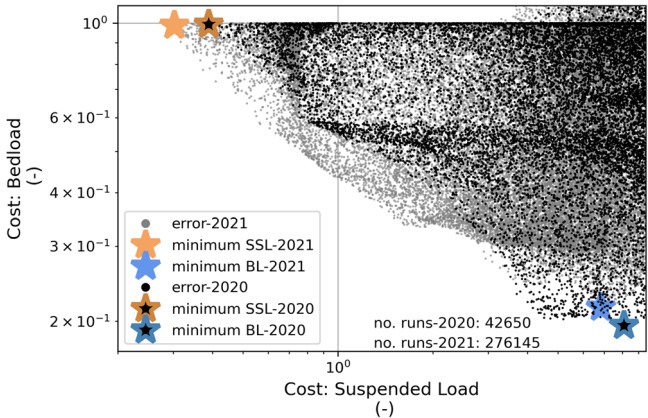

**Fig. 2 | Cost of identical parameter combinations applied to bedload (*BL*) and suspended load (*SSL*) for 2020 and 2021.** Cost is defined as relative error or $\xi(\theta) = \frac{1}{n_r} \sum |\frac{\overline{Q}(\theta) - \overline{Q}_o}{\overline{Q}_o}|$, where $n_r$ is the number of runs, $\theta$ is a parameter set, $\overline{Q}(\theta)$ is model output and $\overline{Q}_o$ is observations. Note that no parameter combinations result in model outputs that perform well for both *BL* and *SSL*.

optimum parameter combinations, the model's mean relative error is between 8 % and 20 % for both *BL* and *SSL* cases. Furthermore, the model captures observed variations in sediment transport from the glacier (Fig. 3).

The variance of the optimum parameter combinations amongst the experiments further highlights the different processes that control *BL* and *SSL* export (Fig. 4). These distributions are significantly different across all four experiments. Yet, with the exception of sediment production ($\dot{\varepsilon}$) and friction factor ($f_i$) each parameter correlation that is positively or negatively correlated, or insignificant, in *SSL* 2020 remains so in 2021 (Fig. 4a, c). This trend between years suggests that similar hydro-geomorphic processes controls *SSL* export. For instance, sediment production ($\dot{\varepsilon}$) and initial till thickness condition ($H_0$) are negatively correlated in both years, as are channel shape factor ($\beta$) and initial till thickness condition ($H_0$). Fewer such parameter interactions occur amongst the other three cases (Fig. 4). This agreement between 2020 and 2021 highlights consistency in SSL export processes between the 2 years and the model's ability to represent them.

In the *BL* case, the hydraulic parameters channel shape factor ($\beta$) and friction factor ($f_i$) are negatively correlated in both 2020 and 2021. However, other similar interactions between the 2 years do not emerge. This absence could be due in part to the relatively short study period in 2020 that reduced the sensitivity of the sediment availability parameters, sediment production ($\dot{\varepsilon}$) and initial till thickness condition ($H_0$).

**Fig. 3 | Model outputs from the four numerical experiments (suspended load (*SSL*) and bedload (*BL*), 2020 and 2021).** **a–j** water discharge from the two seasons. **b–k** Sediment transport capacity (blue) and sediment discharge (red) for the cases, shaded areas represent range of outcomes. **c–l** Observations ($\overline{Q}_o$; orange) and range of accepted model outputs ($\overline{Q}$; blue). Note that parameter inversion for *BL* in 2020 was not successful, hence the poor model performance.

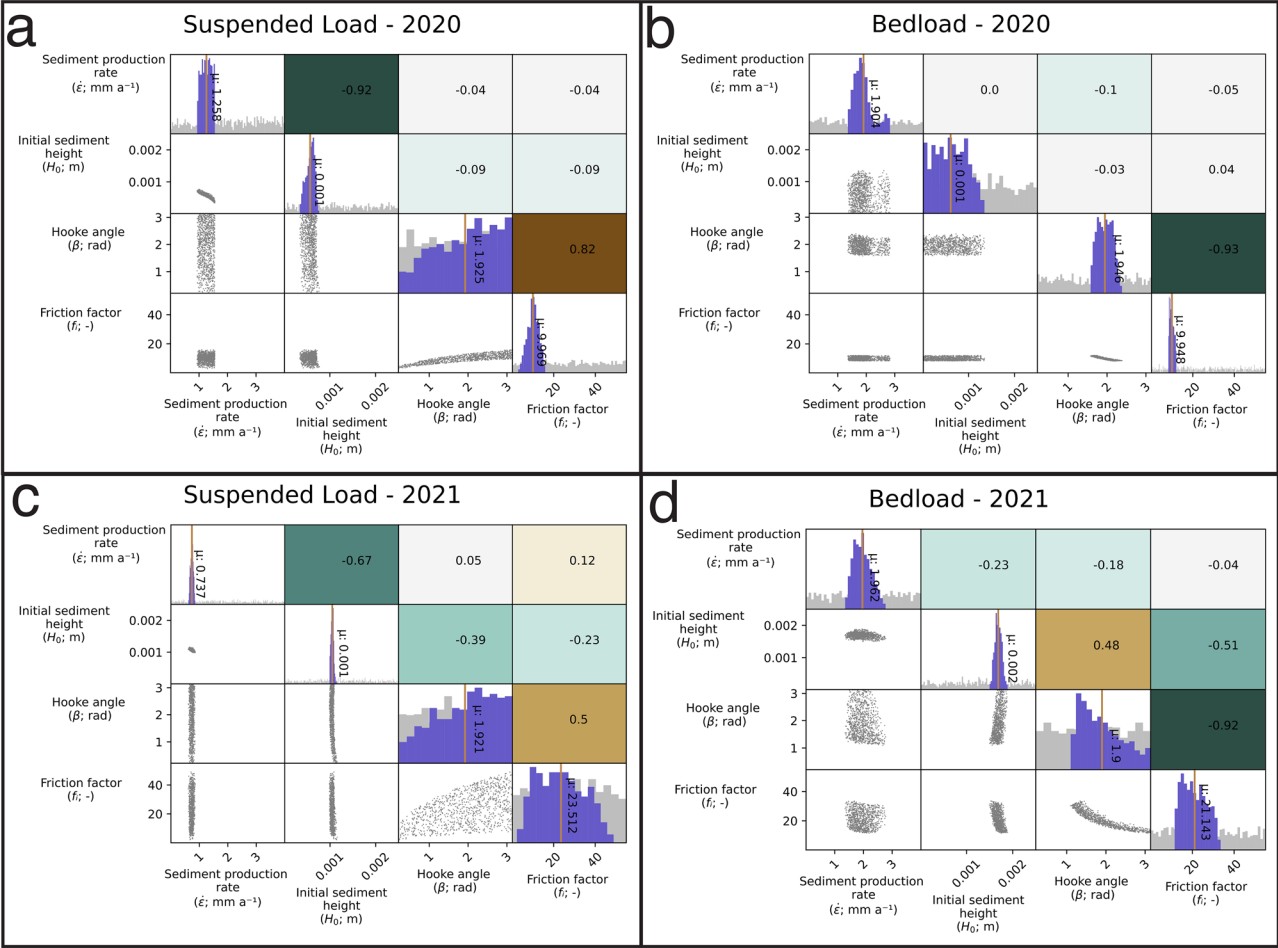

**Fig. 4 | Correlation plots for the four numerical experiments (suspended load (SSL) and bedload (BL), 2020 and 2021).** Note a similar trend in correlations between suspended load in 2020 (**a**) and 2021 (**c**). Parameter correlations for bedload experiments given in (**b**) and (**d**). Red vertical lines show mean parameter values. Posterior histograms are in purple on diagonal, with the uniform prior distributions denoted by gray histograms. Green (brown) colors denote negative (positive) Pearson's correlation value of the corresponding scatter plot. Boxes filled with gray denote insignificant correlation at the 95% confidence interval.

## Processes controlling the export of bedload and suspended load

Forward model outputs and inverted parameters demonstrate the different sediment availability and mobilization processes driving *SSL* and *BL* export (Figs. 2, 4). To establish the particular processes controlling sediment export, we examine *SSL* and *BL* results from both years. However, we note that the 2020 model runs cover a substantially shorter time period, because the model is not able to capture a severe reduction in *BL* export observed this season[33,44] (see Methods; Figs. S4–S5). Forward model outputs show that sediment discharge capacity far exceeds sediment discharge in the *SSL* cases (Fig. 3b, h). Excess *SSL* sediment transport capacity (i.e. suspended sediment load sediment transport capacity) suggests that *SSL* operates in a supply-limited regime. However, a transport-limited regime controls *BL* export at times, with small differences between sediment discharge capacity and sediment discharge (Fig. 3k). Due to the supply-limited *SSL* regime, the parameters controlling subglacial sediment availability develop narrow posterior parameter distributions (sediment production rates $\dot{\varepsilon}$) and initial sediment thickness conditions ($H_0$; Fig. 4a, c).

A negative correlation occurs between sediment production rate ($\dot{\varepsilon}$) and initial sediment thickness condition ($H_0$) for *SSL* in both years (Fig. 4). The relationship develops as lower erosion rates must compensate for higher initial sediment thickness to produce the observed quantity of *SSL*[45]. In the *SSL* case, the posterior distributions for variables driving sediment transport capacity, $\beta$ and $f_i$, do not interact with the erosion rate ($\dot{\varepsilon}$; Fig. 4a). $\beta$ and $f_i$ values span the parameter space,

resulting in viable subglacial water pressure and water velocity conditions (see Methods; Fig. 5). As a result, they do not represent particular water velocity and channel growth conditions that control sediment transport capacity, which would control SSL export in a transport-limited regime (Fig. 6).

Sediment availability controls *BL* export as well in both years, although optimum parameter combinations for *BL* in 2021 are narrower than in 2020. This suggests less influence on sediment availability in 2020, likely due to the significantly shorter study period that year which limits the influence of sediment availability. Here, sediment production rate ($\dot{\varepsilon}$) and initial sediment thickness condition ($H_0$) develop optimum parameter distributions that deviate significantly from their prior ones (Fig. 4a, c). During the longer study period 2021, the negative correlation between sediment production rate ($\dot{\varepsilon}$) and initial sediment thickness condition ($H_0$) shows that a specific quantity of sediment must be available for an optimum model run. Despite the dependence on sediment availability, in *BL* for both 2020 and 2021 a strong negative relationship occurs between the two hydraulic parameters, channel shape ($\beta$) and roughness ($f_i$). This relationship lies contrary to the positive relationship between the $\beta$ and $f_i$ parameters in the *SSL* cases (see Methods, Fig. 5). Here, the optimum hydraulic parameters for *BL* are a subset of those resulting in feasible subglacial hydraulic conditions for Otemma glacier (Figs. 4d, 5). Therefore, the optimum parameter distributions reflect the specific hydrological conditions to capture the observed *BL* export, not possible subglacial hydraulic conditions.

Our model parameterizes *SSL* and *BL* with different relationships (*SSL*: Engelund and Hansen, 1967[42] and *BL*: Wilcock and Crowe, 2003[43]). In turn, we compare hydraulic factors such as water velocity and shear stress to examine the different hydraulic conditions between the *SSL* and *BL* optimum parameter combinations. For 2021, the longer study period with more *BL* variations (Fig. 3d), the subset of optimum hydraulic parameters for the *BL* numerical experiments does not result in a specific mean sediment transport capacity over the season. In *BL*, parameter combinations of channel shapes ($\beta$) and roughness ($f_i$) result in mean seasonal shear stress integrated across the channel bed width, spanning 87 % of the viable hydraulic conditions (Fig. 6d, h). Furthermore, the *BL* parameters result in seasonal mean channel width and water velocities covering a substantial range of viable subglacial hydraulic conditions as well (37 % and 70 % of the viable range, respectively).

The relationship between the channel shape ($\beta$) and roughness ($f_i$) parameters in the *BL* cases for 2021, instead, result in a narrow seasonal mean unit shear stress, encompassing only 11% of the viable subglacial hydraulic conditions (Fig. 6b, f). Increasing channel shear stress would cause both the unit sediment transport capacity to increase and the channel's growth rate to accelerate[e.g.34,46–48]. Over short timescales (i.e. less than a day), increases in water discharge will result in rapid increases in shear stress and thus sediment transport capacity[34,49]. However, at longer timescales, increased shear stress has a

counteracting effect on sediment transport capacity. At time scales longer than days, elevated shear stress causes the subglacial channel size to grow, reducing water velocity and sediment transport capacity[34,49]. In turn, these channel size-water velocity adjustments likely filter high *BL* events in response to high sediment transport conditions. Similar processes occur in subaerial channels[50], but likely over longer periods than the continuous channel size evolution here[34]. Feedbacks between channel growth and sediment transport capacity appear to heavily impact the evolution of sediment discharge in response to water discharge variations and peak discharge events (Fig. 6). As a result, further observations that evaluate subglacial channel shape or water velocity[51] might yield more rigorous parameterizations of both subglacial hydrology and bedload transport. The model outputs from 2020 exhibit a similar trend in these hydraulic factors as compared to 2021, with higher sensitivity to shear stress compared to other factors (Fig. 6a–h).

## Discussion

The above results demonstrate that different processes control *BL* and *SSL* export, with *BL* being partially controlled by hydraulic factors and *SSL* fully responding to sediment availability at Otemma glacier. Interplay between hydraulic parameters values for channel shape ($\beta$) and roughness ($f_i$) can result in viable subglacial hydraulic conditions for *BL* export (Fig. 5)[51,52]. However, results here suggest that only a narrow range of parameter combinations can result in the sediment transport capacity variations needed to capture the observed *BL* export (Fig. 4). Other glaciers may exhibit different dependence on sediment production rate ($\dot{\varepsilon}$) and initial sediment thickness ($H_0$) that impacts their sediment export characteristics[e.g.16]. For instance, the strong dependence of *SSL* export on glacier sliding for a relatively steep glacier may suggest reduced sediment storage and greater dependence on sediment production rate ($\dot{\varepsilon}$)[27]. Conversely, thick sediment has been observed below the Greenland Ice Sheet[53], suggesting greater initial sediment thickness condition ($H_0$) there. Additionally, values of sediment production rate ($\dot{\varepsilon}$) and initial sediment thickness condition ($H_0$) could depend on the time period of the inversion. For instance, longer study periods may result in larger initial sediment thickness conditions ($H_0$) to account for the continuity of sediment availability across the inversion period[45].

These findings support previously proposed mechanisms for *SSL* increases under climate warming[9,10,28,30,38]. *SSL* model performance responds strongly to sediment availability both through sediment production ($\dot{\varepsilon}$) and sediment storage ($H_0$; Fig. 4). The suggested increase occurs as rising melt elevations on glaciers leads to the

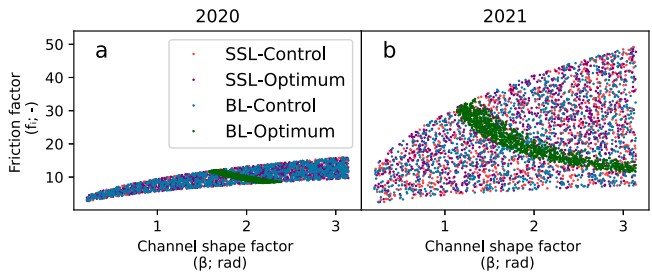

**Fig. 5 | Relationship between $\beta$ and $f_i$ for suspended load (*SSL*) and bedload (*BL*).** Relationship between $\beta$ and $f_i$ for SSL and BL for 2020 (**a**) and 2021 (**b**). Control points denote model runs with realistic hydraulic conditions and not those selected based on model performance. Selected points are a subset of control points with the top-performing model parameters. Notice that both *SSL* and *BL* develop positive correlation in the control points based upon the culling procedure (see Methods). However, a negative relationship emerges in *BL* for optimizing for sediment transport, pointing to a range of specific hydraulic conditions.

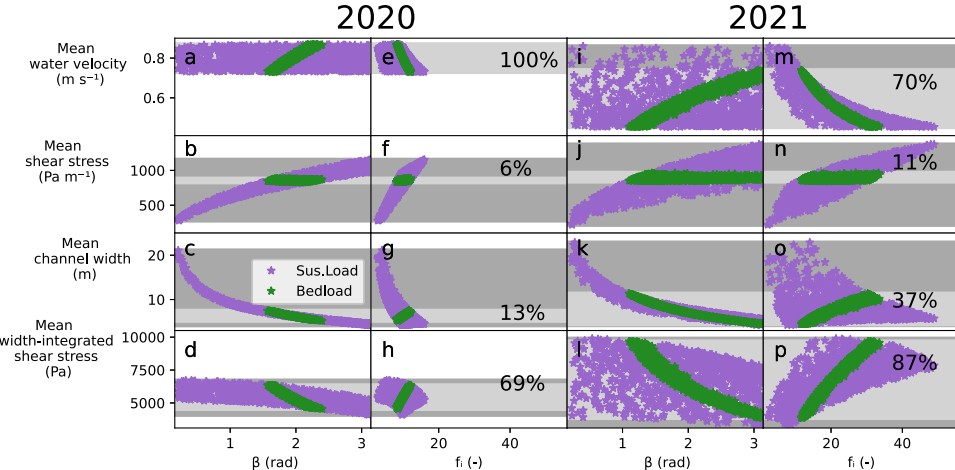

**Fig. 6 | Differences in hydraulics amongst suspended load (*SSL*) and bedload (*BL*), 2020 and 2021.** Water velocity, shear stress, channel width, and width-integrated shear stress averaged over both seasons with respect to channel shape ($\beta$; **a–d, i–l**) and friction ($f_i$; **e–h, m–p**) factors. Light (dark) gray bars denote the range of values for *BL* (*SSL*). The percentages in each row denote the range of *BL* model outputs within the range of *SSL* conditions.

entrainment of previously inaccessible sediment at high elevations, instead of particular subglacial hydraulic conditions that partially impact BL transport. Furthermore, supply-limited transport conditions in *SSL* suggest that little suspended sediment persists in the main channels with high sediment transport capacity (Fig. 3) or at lower elevations of glaciers[30]. Instead, suspended sediment likely originates from higher-order channels with lower water discharge and sediment transport capacity[54].

Similarly, sediment availability's role in representing *SSL* export could support using empirical "erosion rules" in parameterizing glacier abrasion and suspended sediment export[e.g.6,27,37,55,56]. "Erosion rules" could well represent sediment production through abrasion, itself heavily dependent on glacier sliding and particularly abrasive power at the glacier bed[2,11,27,37,57]. *SSL* numerical experiments here suggest that sediment is evacuated with relative ease once produced by abrasion and accessed by a subglacial channel, independent of subglacial sediment transport conditions (Fig. 3b,h). Furthermore, "erosion rules" are largely based on suspended sediment measurements[7,58] (i.e. turbidity measurements, remote sensing of sediment plumes, or sediment cores). As these relationships represent glacier abrasion and suspended sediment export well, they could be referred to as "abrasion rules."

*BL*, on the other hand, is evacuated less efficiently (Fig. 3), potentially leaving substantial sediment available at the glacier bed. *BL*'s partial dependence on sediment transport availability could be even less than the analysis suggests here (Fig. 4). The model's lumped nature means it over-estimates sediment access to subglacial channels, and thus evacuation efficiency[45]. Furthermore, additional subglacial channels underneath the glacier, beyond the single one in this model, could significantly reduce subglacial sediment transport capacity[15]. As a result, the tendency for bedload sized sediment to be retained or deposited underneath glaciers may cause it to insulate bedrock from glacial erosion by quarrying or abrasion[39,41,59-61]. Therefore, glacial erosion of bedrock may only occur if sediment is evacuated by non-fluvial means such as entrainment by the ice[e.g.62] or sediment deformation[e.g.63].

*BL*'s response to changing glacio-climatic conditions remains uncertain given its dependence on subglacial hydraulic factors. The evolution of subglacial hydraulics as glaciers thin and melt increases results in competing interactions between changing hydrological regimes[e.g.64], increasing water discharge variability[65] and slowing channel closure rates as ice thins[66]. Additionally, the especially strong sensitivity of glacial quarrying to subglacial hydrology[36,67] makes it challenging to identify conditions of future *BL* sediment production, whereas the changes to glacier sliding could be more straight forward to identify.[68-70]

Our analysis demonstrates that different physical processes control *SSL* and *BL* export from glaciers, despite significant export quantities of each. These differences mean that bedload and suspended sediment transport types should be considered individually when evaluating sediment export and erosion rates from glaciers, similar to rivers[18]. As results suggest sediment availability drives suspended sediment export, using empirical "abrasion/erosion rules" to quantify sediment production can work well for suspended sediment export. However, when evaluating bedload export, particular attention should be paid to the subglacial hydraulic conditions that drive the channel growth and water velocity as they respond to water discharge variations, along with sediment availability. While subglacial hydraulic processes can be difficult to evaluate, they drive bedload export through glacierized catchments, impacting downstream fluvial systems. Furthermore, the retention of sediment at the glacier bed may armor it from further erosion, invoking other processes necessary to evacuate sediment and maintain bedrock erosion.

## Methods

### Datasets
The model requires inputs of water discharge and glacier topography to be compared with bedload and suspended sediment discharge from the glacier.

We use water discharge, bedload discharge, and suspended sediment discharge data collected from Otemma glacier (45° 57′ 34″ N, 7° 27′ 21″ E) during summers of 2020 and 2021 at a 2 min resolution[33]. The continuous record of suspended sediment export was evaluated using turbidity probes calibrated to a turbidity-suspended sediment relationship. The flux of suspended sediment was established by multiplying the suspended sediment concentration by the coincident water discharge, also measured at the site using a stage height-water discharge relationship. Bedload transport flux was monitored using seismic data. To quantify the bedload flux, seismic data was post-processed using a geophysical inversion model[71] in the open source R package *eseis* (version 0.5. 0). Field tests were used to establish model parameters and the model was calibrated using coincident water stage measurements as no direct measurements of bedload transport were available[33]. We also note, that temporal changes in bedload export also correspond to changing behavior of the proglacial river identified through imagery analysis[44]. These datasets are introduced to the model using a linear interpolated spline, and timestamps with no data were omitted from the analysis.

Ice thickness in the main trunk of Otemma glacier was estimated to be 150 m using available data[72]. The glacier is estimated to be 6 km long and on average 1 km wide based upon 2020 aerial photographs from SwissTopo(https://map.geo.admin.ch). Otemma glacier's terminus is at 2496 ma.s.l., while we consider its maximum elevation of the main glacier at Col de Charmotane 3016 ma.s.l.

### Subglacial sediment transport model
We use a lumped-element model that evolves the thickness of a sub-glacial sediment layer in response to transport and supply-limited sediment transport conditions and erosional processes[45]. A lumped subglacial hydraulics model was used to establish sediment transport capacity[34].

A lumped-element model simulates the evolution of the thickness of a subglacial till layer $H$[45]. Sediment production, or bedrock erosion, adds sediment to the layer. Fluvial sediment transport in supply- and transport-limited regimes mobilizes or deposits sediment, thus removing or adding material to the till layer, respectively[16,73]. The model uses the Exner Equation[74-76], a mass conservation relationship, to evolve the till layer height

$$\frac{\partial H}{\partial t} = -\frac{Q_s}{l\,w} + \dot{m}, \tag{1}$$

where $H$ is the till layer height, $l$ ($w$) is the glaciers' length (width), $\dot{m}$ is a sediment production rate. $Q_s$ represents fluvial sediment flux underneath the glacier

$$Q_s = \begin{cases} \widehat{Q} & \text{if } \widehat{Q} \le \dot{m}\,l\,w & (2a) \\ \widehat{Q}\,\sigma(H) + \dot{m}\,w\,(1-\sigma(H)) & \text{otherwise,} & (2b) \end{cases} \tag{2}$$

$\widehat{Q}$ is the sediment discharge capacity (see equations (9) and (10)), $\sigma$ is a sigmoidal function, smoothing the transition from transport to supply-limited conditions when $H$ is small, effectively reducing sediment availability[16].

The source term $\dot{m}$ is defined as,

$$\dot{m} = \dot{\varepsilon}\,\max\left(0, 1-\frac{H}{H_{\max}}\right), \tag{3}$$

where $H_{max}$ denotes the maximum height of till beyond which sediment production ceases and $\dot{\varepsilon}$ represents the rate of bedrock erosion held steady through the model run (Table S1). In the application here, erosion rate is an explicit parameter[45], unlike previous work[16,54], where erosion depends on glacier sliding or shear stress.

We leverage a lumped R-channel model[34] to parameterize the subglacial hydraulics and sediment transport capacity needed to establish $Q_s$ (Equation (1)). This model assumes that water flows through a subglacial channel with length $l$, underneath a glacier a mean ice thickness of $h_{ice}$. Channel size evolves $S$ with an opening term representing frictional heating from water flow and a closure term from ice creep

$$\frac{\partial S}{\partial t} = C_1 \frac{Q_w \Delta h}{l} - C_2 \left(h_o - \overline{h}\right)^n S, \tag{4}$$

where $t$ represents time, $C_1 = (1 - \rho_w c_p c_t)\frac{\rho_w g}{\rho_i L}$ and $C_2 = 2A\left(\frac{\rho_w g}{n}\right)^n$ are constants (refer to Table S1), $g$ denotes gravitational acceleration, $Q_w$ is the water discharge, $\overline{h} = \frac{1}{2}(h_{ice} + h_p)$ is the mean hydraulic head, with $h_p$ is the proglacial head (0), $l$, $h_o = \frac{\rho_i}{\rho_w}h_{ice}$ indicates the mean ice overburden pressure, $\rho_w$ and $\rho_i$ are the densities of water and ice respectively, and $n$ is Glen's parameter, typically $n = 3$[77].

The head drop $\Delta h$ is

$$\Delta h = l \frac{1}{2g} f_i \frac{v^2}{D_h}, \tag{5}$$

where $f_i$ is a friction factor, $D_h$ is the hydraulic diameter, $l$ is the channel or glacier length, and $v = \frac{Q_w}{S}$ is the water velocity. Channel area $S$ comes from the hydraulic diameter $D_h$ with

$$S = \frac{D_h^2}{2} \frac{\left(\frac{\beta}{2} + \sin\frac{\beta}{2}\right)^2}{\beta - \sin\beta}, \tag{6}$$

where $\beta$ is the central angle of the circular segment of the channel boundary (Hooke angle)[78]. The width of the channel floor $w$ is represented as

$$\widehat{w} = 2\sin\frac{\beta}{2}\sqrt{\frac{2S}{\beta - \sin\beta}}. \tag{7}$$

Shear stress $\tau$, needed in most sediment transport relationships, is established through the Darcy-Weisbach formulation

$$\tau = \frac{1}{8} f_i \rho_w v^2, \tag{8}$$

where $f_i$ is the Darcy-Weisbach friction factor and velocity $v = \frac{Q_w}{S}$. In applying the model to measurements of $SSL$, we implement the total sediment transport relationship of Engelund and Hansen[42]

$$\widehat{Q}_s = \frac{0.4}{f_i} \frac{1}{D_{50}(\frac{\rho_b}{\rho_w} - 1)^2 g^2} \left(\frac{\tau}{\rho_w}\right)^{\frac{5}{2}} \widehat{w}, \tag{9}$$

where $\rho_b$ and $\rho_w$ are the density of the bedrock or sediment, $f_i$ is the friction factor above, $D_{50}$ is median sediment size, $\tau$ is the shear stress evaluated in Equation (8), and $\widehat{w}$ denotes the width of the channel floor that integrates the sediment transport rate across the width of the subglacial channel (Equation (7); Table S1).

To evaluate volumetric bedload transport when comparing model outputs to the $BL$ record, we implement the Wilcock and Crow[43]

sediment transport relationship

$$\widehat{Q}_b = \rho_b W^* \frac{(gR_h\Psi)^{\frac{3}{2}}}{g\left(\frac{\rho_b}{\rho_w} - 1\right)} \widehat{w}, \tag{10}$$

where $R_h$ is the hydraulic radius of the channel ($R_h = \frac{D_h}{4}$), and $W^*$ is the dimensionless transport rate, and $\Psi$ is the hydraulic gradient $\frac{\Delta h}{l}$.

$$W^* = \begin{cases} 0.002\,\tau^{*16.1} & \text{if } \tau^* \leq 1.143 & \text{(11a)} \\ 14\left(1 - \left(\frac{0.85}{\tau^{*0.7}}\right)\right)^{4.5} & \text{otherwise,} & \text{(11b)} \end{cases} \tag{11}$$

where $\psi^*$ is the ratio between the dimensionless bed shear stress and the dimensionless reference bed shear stress, given as

$$\Psi^* = \frac{0.56\,R_h\,\Psi^{\frac{3}{2}}}{D_{50}(\frac{\rho_b}{\rho_w} - 1)}. \tag{12}$$

Note that this relationship is valid for values of $D_{50}$ >4 mm.

## Numerical experiments

We apply the model to records of suspended sediment load ($SSL$) and bedload ($BL$) discharge from the Otemma glacier (Section), using the respective sediment transport relationship (Eqs. (9) and (10)).

We run the model a large number of times with different parameter values and test model outputs against the observed records of sediment discharge in a Monte Carlo framework[45,79]. Parameter values are selected randomly so that they do not depend on the model outcomes of a previous set of parameter. We selected parameter ranges such that distributions formed within their boundaries across all four ensembles ($SSL$ and $BL$ with 2020 and 2021; Fig. figu4). This way parameters from all four ensembles were sampled from the same prior distributions.

To implement this scheme, for each model run we create a vector of randomly selected parameter values $\theta$ to run in the forward model

$$\theta = [\dot{\varepsilon}, H_0, \beta, f_i]. \tag{13}$$

$\dot{\varepsilon}$ is the sediment production rate to produce sediment over the model run used to evaluate the till source $\dot{m}$ in ref. 45. $H_0$ is the initial till height condition used to evaluate the amount of sediment stored below the glacier before the model initialization. $\beta$ is the Hooke angle, controlling the shape and thus the width of the subglacial channel[78]. $fi$ is the Darcy-Weisbach friction factor used to evaluate the water velocity and hydraulic potential to calculate shear stress and evolution of channel size (see ref. 34). The range of possible values and their sampling distribution are given in Table S1.

When running the model forward, we apply a spin-up to establish the initial channel cross-sectional area by applying maximum water discharge over the first 3 days of the study period until the change in cross-sectional area is negligible.

Model outputs ($Q$) with different parameters ($\theta$) are compared against the observed sediment discharge records ($Q_o$) from the $BL$ or $SSL$ datasets using relative error ($\xi$)

$$\xi(\theta) = \frac{1}{n_r} \sum \left|\frac{\overline{Q}(\theta) - \overline{Q_o}}{\overline{Q_o}}\right|. \tag{14}$$

$n_r$ is the number of data points. $\overline{Q}$ and $\overline{Q_o}$ are values averaged over 5 day periods, to reduce the impact of sediment transport times compared to water discharge within the glacier, which is not accounted for in the model e.g. Ref. 80.

For the year 2020, model runs commenced for on June 26, 2020 and ceased on August 31, 2020. For 2021, model runs commenced on June 15, 2021 and ceased on August 1, 2021. These periods coincides

with available bedload and suspended load data. In 2021, data were collected until August 22, 2021. However, a substantial reduction in sediment export from Otemma glacier occurred between August 1 to 3, 2021. The possibility that this resulted from instrument error is minimal as imagery shows that proglacial forefield morphodynamics were tightly coupled to the reduced sediment input[44]. The lumped element model here does not able to capture this transition to reduced sediment export (Fig. S4–S5). Therefore, we only model the beginning of the season in 2020.

We cull model runs if they experience a flotation fraction of > 1.2 or if their hydraulic head exceeds the maximum glacier elevation for >1% of the study period. Additionally, we remove model runs experiencing water velocity > 1.5 m s$^{-1}$.

To test the viability of similar processes to capture *SSL* and *BL*, the model is first run with 500, 000 identical parameter values for *SSL* and *BL* cases over both 2020 and 2021. The comparison is done using Equation (14), with physically infeasible runs being culled.

Additionally, we identify optimum parameter combinations using inversion a Monte Carlo inversion. We run the model with $2 \times 10^6$ randomly selected parameter values and quantify the relative error for each viable one ($\xi$; Equation (14)). For the analysis of both the posterior parameter combinations and their resultant forward model behavior, we utilize the top performing 0.05% (1000) of all runs.

## Data availability
Sediment transport data used in this study is available from Mancini et al.[33].

## Code availability
Running and plotting scripts are available at https://bitbucket.org/IanDelaney/otemma/src/master/. The complete code is available at https://doi.org/10.5281/zenodo.15227775.

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

## Acknowledgements

Funding from Swiss National Science Foundation project 200021-188734/1 (S.N.L.) supported the data collection and supported M.J.

Swiss National Science Foundation project PZOOP2_202024 (ID) supported I. Delaney. We are grateful to B. Schaefli for insightful discussions about experiment design. The Swiss Geocomputing Centre, Université de Lausanne, provided computational resources.

## Author contributions

The concept of this study was developed by I.D. and S.N.L. F.L. and I.D. carried out the model runs. Observational data were collected by M.J., D.M. and S.N.L. I.D. developed the methodology and model. I.D. led the writing process. I.D., F.L., M.J., D.M. and S.N.L. contributed to investigation and editing/review of the manuscript.

## Competing interests

The authors declare no competing interests.
