## [Transparent Peer Review file · Nature Communications]

Different geomorphic processes control suspended sediment and bedload export from glaciers

Corresponding Author: Dr Ian Delaney

Version 0:

Reviewer comments:

Reviewer #1

(Remarks to the Author)

The paper entitled “Different geomorphic processes control suspended sediment and bedload export from glaciers” deals with subglacial erosion and sediment export from glaciers. It is based on a subglacial erosion and sediment transport model that has been compared with high-frequency and continuous records of suspended and bedload fluxes collected during two melt seasons in a glacier (Otemma Glacier, Switzerland). Main results show that suspended sediment exports are mainly controlled by sediment supply, while bedload exports are governed by subglacial hydraulic conditions. This allows the authors to emphasize the importance of separating suspended and bedload transport in subglacial sediment export studies.

My general feeling about this paper is mixed. On one side, I find that the idea of applying a lumped modelling approach to the unique sediment transport dataset collected in the Otemma glacier is quite original, and it allows the authors to bring new insights about processes controlling sediment exports. On the other side, I do not find that the authors are quite convincing in showing how the modelling approach allow them to provide good evidences of these controlling factors. This may be partly due to my low level of expertise in erosion and sediment transport modelling. However, I still believe that the manuscript needs a substantial improvement to better explain how geomorphic processes have been inferred from modelling results. I notably think that the section entitled “Processes controlling the export of bedload and suspended load” is a key part of the paper that has been poorly written, and that fails to provide clear evidences supporting conclusions about controlling factors. The analytical approach must be better explained and justified here.

Another major concern is about sediment transport capacity. This has been computed with the Engelund-Hansen formulae for suspended sediment, and with the Wilcock-Crowe formulae for bedload. It would have been interesting to provide rationales for these choices, because many equations are available, and none of them is considered as optimal in any circumstances. I am notably wondering if similar conclusions would have been obtained if other equations would have been used.

Other comments

Fig. 1a: I do not find these cartoons of subglacial processes graphically efficient. It is particularly true for the parameters represented in the cross-section sketch, which are quite difficult to figure out. A better drawing work would be quite useful for improving the theoretical and graphical impact of this figure. Symbols that appear in these sketches must be explained in the caption.

Fig. 2: cost of parameter combinations that is represented here is unclear to me. This needs to be mathematically explained in the supplementary material.

“As comparable quantities of SSL and BL are exported from the glacier, this suggests that different parameterizations of the two processes are needed”: I do not understand this argument that should be better explained.

“posterior parameter distributions”: needs to be explained in the main text. Does this correspond to model outputs, or to the optimum combination or parameters for BL and SSL?

Typo and wording: see annotated pdf files (article and supplements)

The lack of line numbers does not facilitate the review work...

(Remarks on code availability)

Reviewer #2

(Remarks to the Author)

Review for NCOMMS-24-66300-T

The authors examine the distinct geomorphic processes governing fluvial sediment transport from glaciers, distinguishing between fine-grained suspended sediment and coarser bedload transport. They invert a physically based numerical model for sediment production and transport using a continuous record of bedload and suspended sediment flux. Since both the model architecture and observational data stem from prior work, the primary contribution here lies in synthesizing the two to identify specific controls on each transport mode. Their findings show that while both suspended sediment and bedload export are influenced by sediment availability, bedload transport is further constrained by hydraulic factors such as channel shape and roughness. This highlights that subglacial fluvial systems are more efficient at evacuating suspended sediments, whereas bedload transport requires higher shear stress for mobilization.

The authors conclude that hydraulics should be explicitly considered when examining bedload transport processes and emphasize the need to treat suspended and bedload transport separately in glacial sediment export models. Overall, I find this paper to be generally well-written and of sufficiently broad interest to the community to warrant publication in Nature Communications. Notably, while many previous studies have inferred erosion rates through parameterizations linking suspended sediment in proglacial rivers to glacier dynamics, this study conclusively demonstrates that such an approach captures only a fraction of the total picture. Below, I present my thoughts, first outlining my primary issues, followed by minor concerns and a few grammatical suggestions and rewordings.

Primary issues

My main concern with this work is its heavy reliance on prior publications for key details, making aspects of the study difficult to interpret in isolation. This project builds upon previous papers that explore both the model development and the SSL/BL dataset. I had to search through the referenced papers to locate essential details needed to fully understand the authors' interpretations. For example, of the four main parameters used for the inversion— ϵ' (bedrock erosion potential), H_0 (initial till height condition), β (Hooke angle), and f_i (Darcy-Weisbach friction factor)—none are explicitly found in the equations provided in the methods section, making the analysis feel somewhat opaque. While I appreciate the need for brevity, including a summary of key model elements in the supplementary materials or fleshing out the methods could significantly improve clarity.

- What the justifications are for the range in parameter values? For instance, in Delaney et al. 2024, the range for H_0 and H_f0 is ~10-4 m to 5 cm, whereas in this paper study H_0 ranges from 10-4 to 2 mm. Why the order of magnitude difference here? Similarly, is the increase in ϵ' compared to Delaney et al., 2024 meant to capture the production from abrasion and quarrying versus simply abrasion/SSL, or is there another motivation?
- The study relies on bedload measurements from seismic techniques and suspended sediment from direct measurements, acquired in a previous study. While these methods are reasonable, a more explicit statement of their respective uncertainties and potential biases could be added in this paper, given the poor performance of the BL inversion in 2020.
- "We elucidate the factors controlling sediment export by inverting a physically based numerical model of subglacial sediment production and sediment transport with suspended sediment and the first continuous bedload discharge records from an Alpine glacier." It is not clear to me how the sediment production component in this model is "physically based". Going back to Delaney et al., 2019, I see erosion rate is parameterized as a linear relation with slip speed, with reference to Herman et al., 2015 season (empirical relationship rather than based on first principals). However, this dataset focused on SSL in a proglacial river over a short melt season. Is bedrock erosion rate in this current study just held to a constant sediment production determined through inversion, or does it vary in time with slip speed, etc. ?

The bedload component of 2020 performs poorly with the inversion which the authors attribute to "the lumped-element model's limited ability to capture decreasing BL availability observed from August in the 2020." Additional clarification would be useful here. First, is there speculation as to why this drop-off occurs? The bedload flux is itself the product of an inversion. Are there potential issues with the source data? Further, if the inversion is not designed to capture the decrease, could the analysis be successfully conducted on the data prior to August 2020 instead of the full season to provide at least a partial baseline for comparison against the other BL and SSL parameters?

There is no discussion of how well the best fit parameters found through inversion align with physical expectations for this glacier.

Minor points

“from the Otemma glacier in Switzerland” It would be helpful to briefly state what is known about the bed conditions here. Hard, soft, mixed, unknown, etc.

“Sediment is exported fluviially either as finer sediment in suspension or as coarser bedload with intermittent contact between sediment and the bed.”

“Here, we are able for the first time to show that different processes control bedload and suspended load export from glaciers.” Perhaps “show quantitatively” or “show how.” Other studies have reported similar results suggesting that BL and SSL operate differently with discharge and drainage system evolution (e.g. reference 31)

“The disparity in SSL and BL performance (Figure 2) occurs despite the model’s ability to represent SSL and BL well, individually in three out of four datasets. (Figure 3).”

“...suggesting that similar hydrogeomorphic processes controls SSL export in the two years.” Is the assumption that hydrogeomorphic controls on BL differed between the two years or that the poor performance of the inversion in 2020 precludes a comparison?

“SSL experiments here suggest that sediment is evacuated with relative ease once produced, independent of subglacial sediment transport conditions... BL is evacuated less efficiently (Figure 3), potentially leaving substantial sediment available at the glacier bed.” I have difficulty reconciling this statement with myriad observations of recently exposed or in-situ poorly sorted till deposits. While SSL may be easily mobilized along one section of the bed or in a channel, there are many glaciers with full or partial till cover, which contain an abundance of fines, implying sand-sized and smaller particles are regularly created through comminution and stored in the subglacial environment intermixed with BL-sized particles. Perhaps specify: can be evacuated with relative ease once produced through abrasion.

Minor suggested rewordings:

“To date, the difficulty in observing subglacial bedload transport resulted in limited understanding of the physical processes associated with evacuating bedload compared with suspended load.”

Underneath Beneath glaciers, sediment export

Water transports sediment fluviially either in suspension, composed largely of sand and finer particles, or as bedload, where clasts roll, slide, or saltate over the channel’s bed.

...models suggest that glacier quarrying creates bedload-sized material in amounts comparable to the suspended sediment produced by glacier abrasion.

SSL sediment transport capacity (i.e. suspended sediment load sediment transport capacity)

instead of particular subglacial hydraulic conditions that partially impact BL transport

As a result of the number of subglacial channels and their non-linear effect on transport capacity, BL dependence on sediment transport capacity could be even less than suggested by the analysis here.

Figure 5 y-axis: pa Pa

Methods: Parameter values are selected randomly so that they do not depend dependent on the model outcomes of a previous set of parameter

(Remarks on code availability)

Version 1:

Reviewer comments:

Reviewer #1

(Remarks to the Author)

The revised version of the manuscript is of excellent quality, and the authors have very well addressed all my comments. They also have provided clear and convincing arguments about the changes made to the manuscript. I would therefore recommend acceptance of publication.

After a careful reading of the manuscript and supplementary material, only few typos have been noticed:

L171: delete “this”

L218-219: unclear sentence

Fig. 3: the thick black line separating the two years could be removed

Congratulations to the authors for this very impressive work that should be of great interest to scientists working on glacier erosion and sediment transport

(Remarks on code availability)

Reviewer #2

(Remarks to the Author)

Please refer to attached document with my comments.

(Remarks on code availability)

Dear Editor,

We thank Reviewer 1 for their detailed and constructive comments. Their work, we believe, has provided feedback that has greatly improved the manuscript.

The reviewer's comments are in gray, our response is in black and quotations from the new text are in bold font.

Best regards,

Ian Delaney on behalf of all authors

General Comments

- My general feeling about this paper is mixed. On one side, I find that the idea of applying a lumped modelling approach to the unique sediment transport dataset collected in the Otemma glacier is quite original, and it allows the authors to bring new insights about processes controlling sediment exports. On the other side, I do not find that the authors are quite convincing in showing how the modelling approach allow them to provide good evidences of these controlling factors. This may be partly due to my low level of expertise in erosion and sediment transport modelling. However, I still believe that the manuscript needs a substantial improvement to better explain how geomorphic processes have been inferred from modelling results. I notably think that the section entitled "Processes controlling the export of bedload and suspended load" is a key part of the paper that has been poorly written, and that fails to provide clear evidences supporting conclusions about controlling factors. The analytical approach must be better explained and justified here.

We thank the reviewer for the careful and balanced consideration of our work. While the reviewer points to several positive points, we also take seriously the comments of uncertainty in the modeling approach and the need to better describe the "Processes controlling the export of bedload and suspended load." Following comments from R2, we believe that part of the uncertainty is due to a streamlined and minimal methods explanation. Rather than referencing the methods used, as in the previous version, we have summarized the models and datasets. Furthermore, we have referenced these materials in the main text.

We have carefully considered the ways to improve the "Processes controlling the export of bedload and suspended load" section. Figure S4, formally in the supplement, is now in the text. With the as new model runs have been added, the text has been updated substantially. Additionally, the recommendations made in the attached pdf file have been integrated. We believe that these changes have improved this section.

- Another major concern is about sediment transport capacity. This has been computed with the Engelund-Hansen formulae for suspended sediment, and with the Wilcock-Crowe formulae for bedload. It would have been interesting to provide rationales for these choices, because many equations are available, and none of them is considered as optimal in any circumstances. I am notably wondering if similar conclusions would have been obtained if other equations would have been used.

We thank the reviewers for these comments. Engelund and Hansen (1967) is an equation that is commonly accepted for transport of sand to gravel sized sediment. It has been used extensively and cited over 2500 times (Google Scholar). We have chosen this formula given its validation over sand size particles that are characteristic of subglacial and proglacial streams.

Wilcock and Crowe (2003) is a commonly accepted bedload transport parameterization that has been used extensively as well (1100 citations; Google Scholar). We are aware of no other transport formula that is more commonly accepted for large bedload sized material.

Excess shear stress approaches can be traced back to the 1930s (Shields, 1936) were generalized to gravel in the classic work of Meyer-Peter Müller in the 1940s and then have developed into one of the two main approaches for sediment transport modelling (the other one being stream power based). We might expect the detailed predictions to change if different models were used but not the general patterns. We note that both formulas respond to shear stress in the subglacial channel (now described in the supplementary material) and that increased shear stress is needed to mobilize larger sediment, regardless of the transport equation itself. As a result, we see no reason why the general choice of sediment transport relationship would substantially impact the outcome of the study. When the hydraulic or sediment transport conditions are compared to each other (i.e. Figure 5, now Figure 6), we use shear stress, or shear stress across the channel bed. This way the hydraulic conditions can be compared, regardless of the choice in sediment transport formula.

We have added the following sentences to the text in Section 3. **Our model parameterizes SSL and BL with different relationships (SSL : Englund and Hansen, 1967(Englund and Hansen, 1967) and BL : Wilcock and Crowe, 2003(Wilcock and Crowe, 2003)). In turn, we compare hydraulic factors such as water velocity and shear stress to examine the different hydraulic conditions between the SSL and BL optimum parameter combinations.**

Specific Comments

- Fig. 1a: I do not find these cartoons of subglacial processes graphically efficient. It is particularly true for the parameters represented in the cross-section sketch, which are quite difficult to figure out. A better drawing work would be quite useful for improving the theoretical and graphical impact of this figure. Symbols that appear in these sketches must be explained in the caption.

We thank the reviewer for this comment. After speaking with colleagues we are uncertain of ways to substantially improve the figure. However, we have added the following text to the caption to clarify the different parameters. **(sediment production rate ($\dot{\epsilon}$), the sediment height initial condition (H_0), the channel shape (β), and channel roughness (f_i)).**

- Fig. 2: cost of parameter combinations that is represented here is unclear to me. This needs to be mathematically explained in the supplementary material.

Excellent comment. The details are described in the “Methods” section. However, we have added the following text to the figure caption: **Cost is defined as relative error or $\xi(\theta) = \frac{1}{n_r} \sum \left| \frac{\overline{Q}(\theta) - \overline{Q}_o}{\overline{Q}_o} \right|$, where θ is a parameter set, $\overline{Q}(\theta)$ is model output and \overline{Q}_o is observations.**

- “As comparable quantities of SSL and BL are exported from the glacier, this suggests that different parameterizations of the two processes are needed”: I do not understand this argument that should be better explained.

This point should have been more clearly stated. We have slightly modified the sentence and now believe the intended meaning is clearer. **As substantial quantities of SSL and BL are exported from the glacier(Mancini et al., 2023), this suggests that different parameterizations of the two processes are needed to fully capture subglacial sediment export.**

- “posterior parameter distributions”: needs to be explained in the main text. Does this correspond to model outputs, or to the optimum combination of parameters for BL and SSL?

We have reworded the text to: **optimum parameter combinations** for consistency with the paragraph above it .

- Typo and wording: see annotated pdf files (article and supplements)

We thank the reviewer for carefully reading and editing the manuscript. The corrections proposed in these comments have greatly improved the manuscript.

References

Engelund, F. and Hansen, E. (1967). A monograph on sediment transport in alluvial streams. Technical report, Technical University of Denmark, Copenhagen, Denmark.

Mancini, D., Dietze, M., Müller, T., Jenkin, M., Miesen, F., Roncoroni, M., Nicholas, A., and Lane, S. (2023). Filtering of the Signal of Sediment Export From a Glacier by Its Proglacial Forefield. *Geophysical Research Letters*, 50(21):e2023GL106082. e2023GL106082 2023GL106082.

Shields, A. (1936). Anwendung der Aehnlichkeitsmechanik und der Turbulenzforschung auf die Geschiebebewegung. *PhD Thesis Technical University Berlin*.

Wilcock, P. and Crowe, J. (2003). Surface-based transport model for mixed-size sediment. *Journal of Hydraulic Engineering*, 129(2):120–128.

Dear Editor,

We thank Reviewer 2 for their detailed and constructive comments. Their work, we believe, has provided feedback that has greatly improved the manuscript.

The reviewer's comments are in bold, our response is in italics and quotations from the new text are in normal font.

Best regards,

Ian Delaney on behalf of all authors

General Comments

Reviewer comment: My main concern with this work is its heavy reliance on prior publications for key details, making aspects of the study difficult to interpret in isolation. This project builds upon previous papers that explore both the model development and the SSL/BL dataset. I had to search through the referenced papers to locate essential details needed to fully understand the authors' interpretations. For example, of the four main parameters used for the inversion— $\dot{\epsilon}$ (bedrock erosion potential), H_0 (initial till height condition), β (Hooke angle), and f_i (Darcy-Weisbach friction factor)—none are explicitly found in the equations provided in the methods section, making the analysis feel somewhat opaque. While I appreciate the need for brevity, including a summary of key model elements in the supplementary materials or fleshing out the methods could significantly improve clarity.

Authors' response: We thank the reviewer for bringing these issues to light. Along with comments from Reviewer 1, it is clear that our methodology should be better described and this should significantly strengthen the manuscript. We have addressed the itemized issues below. Additionally, we have included a more detailed summary of the bedload flux evaluation (Mancini et al., 2023), along with summarizing the sediment transport modeling components (Delaney et al., 2024b) and sediment capacity and hydraulics variations (Delaney et al., 2024a).

- **Reviewer comment:** What the justifications are for the range in parameter values? For instance, in Delaney et al. 2024, the range for H_{g0} and H_{f0} is 10-4 m to 5 cm, whereas in this paper study H_0 ranges from 10-4 to 2 mm. Why the order of magnitude difference here? Similarly, is the increase in $\dot{\epsilon}$ compared to Delaney et al., 2024 meant to capture the production from abrasion and quarrying versus simply abrasion/SSL, or is there another motivation?

Authors' response: We should have been clearer about this in the original manuscript. The text below states the conditions for evaluating the parameter ranges. With respect the comparison with Delaney et al. (2024b), we note that in this case values of H_{f0} could be quite large as the proglacial area comprised a relatively small proportion of the catchment. As a result, large variation in this parameter could still result in an adequate volume of sediment being expelled from the catchment. We anticipate that in a catchment with a proportionally larger proglacial area, H_{f0} would be smaller or at least more narrowly defined. With respect to the H_{g0} parameter in Delaney et al. (2024b), this could be larger for several reasons. First, would be variations between catchments and glaciers. However, another important reason is that the inversion there spanned seven years continuously. As a result, there is more freedom in the inversion for large stocks of sediment to be present at the beginning of the model run and depleted throughout. In the application here, H_{g0} is roughly 1–2 mm, compared to 5 mm in Delaney et al. (2024b). This likely better reflects the

potential amount of sediment available over a single season as presented here. Due to the lack of water discharge measurements in the spring, fall and winter at Glacier d'Otemma, a continuous inversion was not possible.

In the application here, we discuss how $\dot{\epsilon}$ could represent both abrasion in the *SSL* case and quarrying in the *BL* case. As Delaney et al. (2024b) was published prior to this work here, we propose that the $\dot{\epsilon}$ value in that manuscript would represent abrasion, not quarrying. We note, however, that the multi year inversion presented in Delaney et al. (2024b) likely means that a lower value of $\dot{\epsilon}$ would be found as this rate would be maintained through low transport time periods (i.e. winter), whereas the value $\dot{\epsilon}$ here would be for the summer period where higher abrasion (and to a lesser extent quarrying) rates can persist (Ugelvig et al., 2018). The implications of the multi-seasonal runs and the lack of knowledge about sediment production in low melt conditions is discussed in Section 5.3 of Delaney et al. (2024b).

In response to this comment, the following text has been added to the methods section:

We selected parameter ranges such that distributions formed within their boundaries across all four ensembles (*SSL* and *BL* with 2020 and 2021; Figure 4). This way parameters from all four ensembles were sampled from the same prior distributions.

- **Reviewer comment:** The study relies on bedload measurements from seismic techniques and suspended sediment from direct measurements, acquired in a previous study. While these methods are reasonable, a more explicit statement of their respective uncertainties and potential biases could be added in this paper, given the poor performance of the *BL* inversion in 2020.

Authors' response: We have included the following text in the methods section: **The continuous record of suspended sediment export was evaluated using turbidity probes calibrated to turbidity-suspended sediment relationship. The flux of suspended sediment was established by multiplying the suspended sediment concentration by the coincident water discharge, also measured at the site using a stage high-water discharge relationship. Bedload transport flux was monitored using seismic data. To quantify the flux, seismic data was post-processed using a geophysical inversion model (Dietze et al., 2019) in the open source R package *eseis* (version 0.5.0). Field tests were used to establish model parameters and the model was calibrated using coincident water stage measurements as no direct measurements of bedload transport were available (Mancini et al., 2023). These datasets were introduced to the model using a linear interpolated spline, and timestamps with no data were omitted from the analysis.**

We also discuss the robustness of the measurements below in response to another comment below.

- **Reviewer comment:** “We elucidate the factors controlling sediment export by inverting a physically based numerical model of subglacial sediment production and sediment transport with suspended sediment and the first continuous bedload discharge records from an Alpine glacier.” It is not clear to me how the sediment production component in this model is “physically based”. Going back to Delaney et al., 2019, I see erosion rate is parameterized as a linear relation with slip speed, with reference to Herman et al., 2015 season (empirical relationship rather than based on first principals). However, this dataset focused on *SSL* in a proglacial river over a short melt season. Is bedrock erosion rate in this current study just held to a constant sediment production determined through inversion, or does it vary in time with slip speed, etc. ?

Authors' response: This is an important point that demands clarification. With a more developed methods section we believe that this point will be clearer. Here, we have chosen erosion rate to be a steady value. In previous work, it was necessary for glacier erosion to be dependent on slip speed as we would expect different parts of the glacier to produce different amounts of sediment, hence the value of a spatially distributed model (Delaney et al., 2019, 2023). However, in this case, because conditions are averaged over the bed of the glacier, we believe that a single value of erosion rate is appropriate. This is discussed fully in Section 5.3 of Delaney et al. (2024b). We believe the term physically based is appropriate here given the representation of physical processes, however, we would welcome suggests for another terminology proposed by the reviewer. The text in the methods has also been adjusted to read:

$\dot{\epsilon}$ represents the rate of bedrock erosion held steady through the model run... Here, erosion rate is a parameter, unlike previous work(Delaney et al., 2019, 2023), where erosion depended on glacier sliding or shear stress.

- **Reviewer comment:** The bedload component of 2020 performs poorly with the inversion which the authors attribute to “the lumped-element model’s limited ability to capture decreasing BL availability observed from August in the 2020.” Additional clarification would be useful here. First, is there speculation as to why this drop-off occurs? The bedload flux is itself the product of an inversion. Are there potential issues with the source data? Further, if the inversion is not designed to capture the decrease, could the analysis be successfully conducted on the data prior to August 2020 instead of the full season to provide at least a partial baseline for comparison against the other BL and SSL parameters?

Authors' response: This is an excellent comment. We have discussed several reasons for a potential drop off in bedload export from the glacier. Possible reasons include behavior of an overdeepening or simply sediment exhaustion in the main channel. However, following the drop off in sediment export from the glacier, the proglacial river transitions from braiding behavior to incision on the basis aerial imagery analysis (Mancini et al., 2024). This behavior of the proglacial river is expected given a reduction in sediment supply from the river. As a result, there is strong evidence from both seismic data and imagery that such a drop off occurred, and that it was not the result of measurement error.

We also thank the reviewer for the recommendation to include the inversion for the first part of the season. This has been done and the results are presented in the updated manuscript. The following text has been added to the “Methods” to explain the updated modeling procedure.

For the year 2020, model runs commenced for on June 26, 2020 and ceased on August 31, 2020. For 2021, model runs commenced on June 15, 2021 and ceased on August 1, 2021. These periods coincides with available bedload and suspended load data. In 2021, data were collected until August 22, 2021. However, a substantial drop off in sediment export from Otemma glacier occurred between August 1 to 3, 2021. The possibility that this resulted from instrument error is minimal as imagery also shows the proglacial river response to the reduced sediment input (Mancini et al., 2024). The lumped element model here was not able to capture this transition to reduced sediment export (Figure S4-S5). Therefore, we only modeled the beginning of the season.

- **Reviewer comment:** There is no discussion of how well the best fit parameters found through inversion align with physical expectations for this glacier.

Authors' response: We thank the review for this comment. The following paragraph has been added to the section "Divergent impact on landscape evolution due to different sensitivities to hydraulic conditions and sediment availability":

The above results demonstrate that different processes control *BL* and *SSL* export, with *BL* being partially controlled by hydraulic factors and *SSL* fully responding to sediment availability at Otemma glacier. Interplay between hydraulic parameters values for channel shape (β) and roughness (f_i) can result in viable subglacial hydraulic conditions (Figure S4)(Werder and Funk, 2009; Pohle et al., 2022). However, results here suggest that only a narrow range of parameter combinations can result in the sediment transport capacity variations needed to capture the observed sediment export (Figure 4). Other glaciers may exhibit different dependence on sediment production rate ($\dot{\epsilon}$) and initial sediment thickness condition (H_0). For instance, the strong dependence of *SSL* export on glacier sliding for a relatively steep glacier may suggest reduced sediment storage and greater dependence on sediment production rate ($\dot{\epsilon}$) (Herman et al., 2015). Conversely, thick sediment has been observed below the Greenland Ice Sheet(Walter et al., 2014), suggesting greater initial sediment thickness condition (H_0) there. Additionally, values of sediment production rate ($\dot{\epsilon}$) and initial sediment thickness condition (H_0) could depend on the time period of the inversion. For instance, longer study periods may result in larger initial sediment thickness conditions (H_0) to account for the continuity of sediment availability across the inversion period (Delaney et al., 2024b).

Specific Comments

- **Reviewer comment:** "from the Otemma glacier in Switzerland" It would be helpful to briefly state what is known about the bed conditions here. Hard, soft, mixed, unknown, etc.

Authors' response: Text now reads: **Otemma glacier is a 6 km long glacier with unknown hard or soft bed characteristics, in an Alpine climate.**

- **Reviewer comment:** "Sediment is exported fluvially either as finer sediment in suspension or as coarser bedload with intermittent contact between sediment and the bed."

Authors' response: Text now reads: **Sediment is exported fluvially either as finer sediment in suspension or as coarser bedload with intermittent contact between sediment clasts and the channel's bed.**

- **Reviewer comment:** "Here, we are able for the first time to show that different processes control bedload and suspended load export from glaciers." Perhaps "show quantitatively" or "show how." Other studies have reported similar results suggesting that *BL* and *SSL* operate differently with discharge and drainage system evolution (e.g. reference 31)

Authors' response: The reviewer is correct that this should be clarified. The text now reads: **Here, we are able for first time to quantitatively that different processes control bedload and suspended load export from glaciers, using records collected directly at the glacier terminus.**

We note that Mao et al. (2014), e.g. reference 31, collected these records some distance downstream of a glacier as discussed earlier in our manuscript. As a result, the signal measured by them is likely not a glacial signal (Mancini et al., 2023). We believe that this new phrasing better captures the novelty of the study,

- **Reviewer comment:** “The disparity in SSL and BL performance (Figure 2) occurs despite the model’s ability to represent SSL and BL well, individually in three out of four datasets. (Figure 3).”

Authors’ response: We believe that this phrasing is clearer: **No single set of parameters accurately represents both SSL and BL simultaneously (Figure 2). However, with specific parameter combinations, the model adequately represents SSL and BL individually (Figure 3).**

- **Reviewer comment:** “...suggesting that similar hydrogeomorphic processes controls SSL export in the two years.” Is the assumption that hydrogeomorphic controls on BL differed between the two years or that the poor performance of the inversion in 2020 precludes a comparison?

Authors’ response: The reviewer is correct. We have added the following paragraph to the text. **In the BL case, the hydraulic parameters channel shape factor (β) and friction factor (f_i) are negatively correlated in both 2020 and 2021. However, other similar interactions between the two years do not emerge. This absence could be due in part to the relatively short study period in 2020 that reduced the sensitivity of the sediment availability parameters, sediment production ($\dot{\epsilon}$) and initial till thickness condition (H_0).**

- **Reviewer comment:** “SSL experiments here suggest that sediment is evacuated with relative ease once produced, independent of subglacial sediment transport conditions... BL is evacuated less efficiently (Figure 3), potentially leaving substantial sediment available at the glacier bed.” I have difficulty reconciling this statement with myriad observations of recently exposed or in-situ poorly sorted till deposits. While SSL may be easily mobilized along one section of the bed or in a channel, there are many glaciers with full or partial till cover, which contain an abundance of fines, implying sand-sized and smaller particles are regularly created through comminution and stored in the subglacial environment intermixed with BL-sized particles. Perhaps specify: can be evacuated with relative ease once produced through abrasion.

Authors’ response: Excellent comment. The text now reads: **SSL experiments here suggest that sediment is evacuated with relative ease once produced by abrasion and accessed by a subglacial channel, independent of subglacial sediment transport conditions...**

Minor suggested rewordings:

- **Reviewer comment:** “To date, the difficulty in observing subglacial bedload transport resulted in limited understanding of the physical processes associated with evacuating bedload compared with suspended load.”

Authors’ response: Text now reads: **To date, the difficulty in observing subglacial bedload transport limits the understanding of the physical processes associated with evacuating bedload compared with suspended load.**

- **Reviewer comment:** Underneath Beneath glaciers, sediment export

Authors’ response: Done.

- **Reviewer comment:** Water transports sediment fluviially either in suspension, composed largely of sand and finer particles, or as bedload, where clasts roll, slide, or saltate over the channel’s bed.

Authors' response: Done.

- **Reviewer comment:** ... models suggest that glacier quarrying creates bedload-sized material in amounts comparable to the suspended sediment produced by glacier abrasion.

Authors' response: Done.

- **Reviewer comment:** SSL sediment transport capacity (i.e. suspended sediment load sediment transport capacity)

Authors' response: Done.

- **Reviewer comment:** instead of particular subglacial hydraulic conditions that partially impact BL transport

Authors' response: Done.

- **Reviewer comment:** As a result of the number of subglacial channels and their non-linear effect on transport capacity, BL dependence on sediment transport capacity could be even less than suggested by the analysis here.

Authors' response: We thank the reviewer for the careful read.

- **Reviewer comment:** Figure 5 y-axis: pa to Pa

Authors' response: Fixed

- **Reviewer comment:** Methods: Parameter values are selected randomly so that they do not depend dependent on the model outcomes of a previous set of parameter

Authors' response: Done.

References

- Delaney, I., Anderson, L., and Herman, F. (2023). Modeling the spatially distributed nature of subglacial sediment transport and erosion. *Earth Surface Dynamics*, 11(4):663–680.
- Delaney, I., Tedstone, A., Werder, M. A., and Farinotti, D. (2024a). Subglacial and subaerial fluvial sediment transport capacity respond differently to water discharge variations. *EGU sphere*, pages 1–25.
- Delaney, I., Werder, M., Felix, D., Albayrak, I., Boes, R., and Farinotti, D. (2024b). Controls on Sediment Transport From a Glacierized Catchment in the Swiss Alps Established Through Inverse Modeling of Geomorphic Processes. *Water Resources Research*, 60(4):e2023WR035589.
- Delaney, I., Werder, M. A., and Farinotti, D. (2019). A Numerical Model for Fluvial Transport of Subglacial Sediment. *Journal of Geophysical Research: Earth Surface*, 124(8):2197–2223.
- Dietze, M., Lagarde, S., Halfi, E., Laronne, J., and Turowski, J. (2019). Joint sensing of bedload flux and water depth by seismic data inversion. *Water Resources Research*, 55(11):9892–9904.
- Herman, F., Beyssac, O., Brughelli, M., Lane, S., Leprince, S., Adatte, T., Lin, J. Y. Y., Avouac, J. P., and Cox, S. C. (2015). Erosion by an alpine glacier. *Science*, 350(6257):193–195.
- Mancini, D., Dietze, M., Müller, T., Jenkin, M., Miesen, F., Roncoroni, M., Nicholas, A., and Lane, S. (2023). Filtering of the Signal of Sediment Export From a Glacier by Its Proglacial Forefield. *Geophysical Research Letters*, 50(21):e2023GL106082. e2023GL106082 2023GL106082.
- Mancini, D., Roncoroni, M., Dietze, M., Jenkin, M., Müller, T., Ouvry, B., Miesen, F., Pythoud, Q., Hofmann, M., Lardet, F., Nicholas, A., and Lane, S. (2024). Rates of evacuation of bedload sediment from an alpine glacier control proglacial stream morphodynamics. *Journal of Geophysical Research: Earth Surface*, 129(8):e2024JF007727. e2024JF007727 2024JF007727.

- Mao, L., Dell'Agnese, A., Huincache, C., Penna, D., Engel, M., Niedrist, G., and Comiti, F. (2014). Bedload hysteresis in a glacier-fed mountain river. *Earth Surface Processes and Landforms*, 39(7):964–976.
- Pohle, A., Werder, M. A., Gräff, D., and Farinotti, D. (2022). Characterising englacial R-channels using artificial moulins. *Journal of Glaciology*, 68(271):879–890.
- Ugelvig, S. V., Egholm, D. L., Anderson, R. S., and Iverson, N. R. (2018). Glacial Erosion Driven by Variations in Meltwater Drainage. *Journal of Geophysical Research: Earth Surface*, 123(0):2863–2877.
- Walter, F., Chaput, J., and Lüthi, M. (2014). Thick sediments beneath Greenland's ablation zone and their potential role in future ice sheet dynamics. *Geology*, 42(6):487–490.
- Werder, M. A. and Funk, M. (2009). Dye tracing a jökulhlaup: II. testing a jökulhlaup model against flow speeds inferred from measurements. *Journal of Glaciology*, 55(193):899–908.

Dear Editor,

Again, we wish to express our gratitude for your work in editing this manuscript and accepting it for publication.

The reviewer comments are in gray, response to the reviewer's comments is in normal font, and quotes from the text are in bold.

Best regards,

Ian Delaney, on behalf of all authors

Reviewer 1

The revised version of the manuscript is of excellent quality, and the authors have very well addressed all my comments. They also have provided clear and convincing arguments about the changes made to the manuscript. I would therefore recommend acceptance of publication. After a careful reading of the manuscript and supplementary material, only few typos have been noticed: Congratulations to the authors for this very impressive work that should be of great interest to scientists working on glacier erosion and sediment transport

Specific Comments

- L171: delete "this"

Done.

- L218-219: unclear sentence

Text now reads:

Therefore, glacial erosion of bedrock may only occur if sediment is evacuated by non-fluvial means such as entrainment by the ice ^{e.g.}(Iverson, 1993) or sediment deformation ^{e.g.}(Hansen and Zoet, 2022).

- Fig. 3: the thick black line separating the two years could be removed

Done.

Reviewer 2

Specific Comments

- Line 57: we are able for first time to: for the first time

Changed to:

Here, first time we demonstrate

- Line 81: parameter values combinations to: value

Changed to:

parameter values using inversion

- Line 171: model outputs from 2020 does not exhibit to: do not exhibit

Done.

- Extra spaces throughout and would be worth double checking: e.g.

– Line 80: and BL , the model was to: BL, the

Done.

- Line 92: bedload (BL) and suspended load (SSL) to: (BL) . . . (SSL)
Done.
 - Line 121: (Figure 3 k) to: (Figure 3k)
Done.
 - Line 158: (Figure 6 b, f). to: (Figure 6b, f)
Done.
 - Line 211: BL , to: BL,
Done.
 - Line 212: BL 's to: BL's
Done.
 - Line 219: ice e.g. 62 or sediment deformation e.g. 63
Done.
- Line 202 “Similarly, sediment availability’s role in representing SSL export could support using empirical “erosion rules” in parameterizing glacier abrasion and suspended sediment export. “Erosion rules” could well represent sediment production through abrasion, itself heavily dependent on glacier sliding.” Small point, but a power-based abrasion rule has a stronger physical basis than rules that depend solely on slip speed (eg Hallet, 2011; Hansen et al, 2023 doi: 10.1130/G50673.1)

Excellent comment. We have changed the text to read:

“Erosion rules” could well represent sediment production through abrasion, itself heavily dependent on glacier sliding, particularly abrasive power at the glacier bed (Hallet, 1979; Alley et al., 1997; Herman et al., 2015; Koppes et al., 2015; Hansen et al., 2023).

References

- Alley, R. B., Cuffey, K. M., Evenson, E. B., Strasser, J. C., Lawson, D. E., and Larson, G. J. (1997). How glaciers entrain and transport basal sediment: physical constraints. *Quaternary Science Reviews*, 16(9):1017–1038.
- Hallet, B. (1979). A theoretical model of glacial abrasion. *Journal of Glaciology*, 23(89):39–50.
- Hansen, D. D., Brooks, J. P., Zoet, L. K., Stevens, N. T., Smith, L., Bate, C. E., and Jahnke, B. J. (2023). A power-based abrasion law for use in landscape evolution models. *Geology*, 51(3):273–277.
- Hansen, D. D. and Zoet, L. K. (2022). Characterizing sediment flux of deforming glacier beds. *Journal of Geophysical Research: Earth Surface*, page e2021JF006544.
- Herman, F., Beyssac, O., Brughelli, M., Lane, S., Leprince, S., Adatte, T., Lin, J. Y. Y., Avouac, J. P., and Cox, S. C. (2015). Erosion by an alpine glacier. *Science*, 350(6257):193–195.
- Iverson, N. R. (1993). Regelation of ice through debris at glacier beds: Implications for sediment transport. *Geology*, 21(6):559–562.
- Koppes, M., Hallet, B., Rignot, E., Mougnot, J., Wellner, J. S., and Boldt, K. (2015). Observed latitudinal variations in erosion as a function of glacier dynamics. *Nature*, 526(7571):100–103.

Response NCOMMS-24-66300A

It is my opinion that the authors have addressed the revisions satisfactorily, and I no longer have any outstanding concerns regarding this manuscript. I recognize and appreciate the thoughtful attention they gave to the reviewer feedback and fully support its publication in *Nature Communications*.

During my readthrough I noticed a few small issues, which I detail below:

- Line 57: we are able for first time → for the first time
- Line 81: parameter values combinations → value
- Line 171: model outputs from 2020 does not exhibit → do not exhibit
- Extra spaces throughout and would be worth double checking: e.g.
 - Line 80: and BL , the model was → BL, the
 - Line 92: bedload (BL) and suspended load (SSL) → (BL) ...(SSL)
 - Line 121: (Figure 3 k) → (Figure 3k)
 - Line 158: (Figure 6 b, f). → (Figure 6b, f)
 - Line 211: BL , → BL,
 - Line 212: BL 's → BL 's
 - Line 219: ice ^{e.g. 62} or sediment deformation ^{e.g. 63}.

Line 202 “Similarly, sediment availability’s role in representing SSL export could support using empirical “erosion rules” in parameterizing glacier abrasion and suspended sediment export. “Erosion rules” could well represent sediment production through abrasion, itself heavily dependent on glacier sliding.” *Small point, but a power-based abrasion rule has a stronger physical basis than rules that depend solely on slip speed (eg Hallet, 2011; Hansen et al, 2023 doi: 10.1130/G50673.1)*